# Accuracy and Reliability of Software Navigation for Acetabular Component Placement in THA: An In Vitro Validation Study

**DOI:** 10.3390/medicina58050663

**Published:** 2022-05-14

**Authors:** Alex W. Brady, Jakub Tatka, Lorenzo Fagotti, Bryson R. Kemler, Bradley W. Fossum

**Affiliations:** 1Steadman Philippon Research Institute, Vail, CO 81657, USA; bkemler@sprivail.org (B.R.K.); bfossum@sprivail.org (B.W.F.); 2The Steadman Clinic, Vail, CO 81657, USA; jtatka@sprialumni.org (J.T.); lfagotti@sprialumni.org (L.F.)

**Keywords:** hip, arthroplasty, replacement

## Abstract

*Background and Objectives*: Intraoperative fluoroscopy can be used to increase the accuracy of the acetabular component positioning during total hip arthroplasty. However, given the three-dimensional nature of cup positioning, it can be difficult to accurately assess inclination and anteversion angles based on two-dimensional imaging. The purpose of this study is to validate a novel method for calculating the 3D orientation of the acetabular cup from 2D fluoroscopic imaging. *Materials and Methods*: An acetabular cup was implanted into a radio-opaque pelvis model in nine positions sequentially, and the inclination and anteversion angles were collected in each position using two methods: (1) a coordinate measurement machine (CMM) was used to establish a digitalized anatomical coordinate frame based on pelvic landmarks of the cadaveric specimen, and the 3D position of the cup was then expressed with respect to the anatomical planes; (2) AP radiographic images were collected, and a mathematical formula was utilized to calculate the 3D inclination and anteversion based on the 2D images. The results of each method were compared, and interrater and intrarater reliably of the 2D method were calculated. *Results*: Interrater reliability was excellent, with an interclass correlation coefficient (ICC) of 0.988 (95% CI 0.975–0.994) for anteversion and 0.997 (95% CI 0.991–0.999) for inclination, as was intrarater reliability, with an ICC of 0.995 (95% CI 0.985–0.998) for anteversion and 0.998 (95% CI 0.994–0.999) for inclination. Intermethod accuracy was excellent with an ICC of 0.986 (95% CI: 0.972–0.993) for anteversion and 0.993 (95% CI: 0.989–0.995) for inclination. The Bland–Altman limit of agreement, which represents the error between the 2D and 3D methods, was found to range between 2 to 5 degrees. *Conclusions*: This data validates the proposed methodology to calculate 3D anteversion and inclination angles based on 2D fluoroscopic images to within five degrees. This method can be utilized to improve acetabular component placing intraoperatively and to check component placement postoperatively.

## 1. Introduction

Appropriate acetabular component position regarding inclination and anteversion in total hip arthroplasty (THA) is critical to implant function and longevity. Malposition of the acetabular component may lead to instability, edge loading, impingement, and early failure [1,2,3,4,5]. Surgeons have traditionally relied on physical anatomic and bony landmarks as well as direct intraoperative visualization to achieve appropriate anteversion and inclination of the cup [4,6]. Classic teaching has referenced the “safe zone” of cup inclination and anteversion (40 ± 10° and 15 ± 10°, respectively). Lewinnek et al. described this as a way to minimize dislocation risk after primary THA [4]. However, accurate placement within the safe zone can be challenging using a freehand technique with direct visualization, due to the inherent difficulty of evaluating three-dimensional angles with variable patient alignment on the operating table (more specifically the position and tilt of the pelvis) [7,8]. Callanan et al. have reported that approximately 50% of implants were within an acceptable safe range in conventional THA and hip resurfacing [3]. Further, the Lewinnek safe zone may not be as accurate as previously thought due to the broad range of cup orientations that result in a higher proportion of outliers [4,9]. Thus, a more accurate methodology to intraoperatively assess acetabular cup anteversion and inclination is needed to limit the risk of dislocation, which is the most common complication of THA within the first two years postoperatively [9,10]. 

There are several existing techniques that aim to assess acetabular cup position. Computer navigation was developed in the 1990s, with multiple studies showing a significant improvement in cup positioning [11,12,13,14]. However, the popularity of this technique has decreased due to its complexity and increase in operation time. A mechanical device pinpointing the anterior pelvic plane to optimize acetabular component placement was developed by Kievit et al., who found that it improved positioning with respect to the Lewinnek’s safe zones; however, this device has not been proven to work in the clinical setting [15]. Perioperative imageless techniques use infrared optical stereoscopy that allows for three-dimensional tracking of the prosthetic components and tools. This method does not require a specific patient position or exposure to radiation. However, the associated hardware has excessive costs, and the calibration procedures increase operative time [16]. Finally, recent studies evaluating the use of intraoperative fluoroscopic imaging to optimize component position have shown significant improvements when performing THA [6,17,18,19,20,21]. However, manual acquisition and interpretation of these images present challenges in an intraoperative setting, including variability in the radiographic acquisition technique and complex geometric formulas to assess the 3D position of the cup from the 2D fluoroscopic image obtained [6,17,18,22,23,24,25]. 

The present study proposes a simplified fluoroscopic assessment method that can be used intraoperatively to accurately measure acetabular cup anteversion and inclination. Quantitative radiographic assessment tools utilizing software systems have been previously reported in the postoperative setting [26]. However, validation of a quantitative intraoperative fluoroscopic software system has not been previously described. 

The purpose of this study is to validate the intraoperative software system as an accurate and reproducible method to measure the position of the acetabular component obtained from fluoroscopic images. This method, if validated, could provide surgeons with an intraoperative tool to improve component positioning, better analyze clinical outcomes including dislocation rates and early failures, and redefine the ideal implant position for total hip arthroplasty.

## 2. Materials and Methods

### 2.1. Experimental Setup

A pelvis model (Sawbones USA, Vashon Island, Washington, DC, USA) was securely clamped to a table and aligned with a 9-inch C-arm (OEC, GE Healthcare, Salt Lake City, UT, USA) to produce PA radiographs (Figure 1). Positioning of the pelvis model with respect to the C-arm was performed by aligning the anterior pelvic plane, which is defined as the plane of the two anterior superior iliac spines and the pubic symphysis, to the plane of the X-ray beam, and the cup was positioned in the center of the X-ray beam to eliminate parallax. The radiographic teardrop line was physically marked with small steel beads. This was accomplished manually by a trained orthopedic surgeon, who drilled two small holes in the sawbones model under fluoroscopic guidance. The two small steel beads were then glued into the holes, so the same points could be collected both radiographically and physically using the CMM. A coordinate measuring machine was rigidly clamped to the table and was used to take the 3D measurements. 

### 2.2. Experimental Design

Fluoroscopic PA hip images of an implanted acetabular component (Depuy Pinnacle, Warsaw, IN, USA) were taken with the cup positioned at every combination of 20, 40, and 60 degrees of inclination and 10, 25, and 40 degrees of anteversion, to within 5 degrees of each target position. The nine tested cup positions were chosen to validate the accuracy of the software over a wide range of possible anteversion and inclination angles. Indeed, while it is important for the software solution to accurately assess cup anteversion when the cup is well-placed, it is also important for it to give valid measurements when the cup is incorrectly placed, so the surgeon can appropriately correct it. The range of 20–60 degrees of inclination and 10–40 degrees of anteversion was chosen to encompass the range of positions seen in practice, based on the senior author’s experience in the clinical setting. For each position, X-ray images were taken with the cup alone, with a stem and a small (32 mm) femoral head, and with a stem and a large (36 mm) femoral head. It was expected that the radiographs taken with the femoral head present would have lower accuracy than those taken with the cup alone because it would be more difficult to correctly identify the acetabular rim on the radiographic image with the femoral head in the way. Therefore, both sets were collected to determine if the methodology was viable only in the intraoperative setting (no femoral head) or also in the postoperative setting (with femoral head). The 3D position of the cup was also collected using the CMM in each position, and the results of the two methods were compared. 

### 2.3. 3D Analysis

To compute the true anatomic three-dimensional anteversion and inclination angles, a pelvic coordinate frame was established using a coordinate measuring machine (Romer Absolute Arm, Hexagon Metrology, Wetzlar, Germany). The anterior axis was defined as perpendicular to the plane of the X-ray by collecting six points across the surface of the X-ray receiver and interpolating a best-fit plane through those points, as shown in Figure 2. The lateral axis was defined as the line connecting the two digitized teardrop points. The superior axis was defined as the cross product of the anterior and lateral axes. Finally, the lateral axis was re-calculated as the cross product of the anterior and superior axes to ensure an orthonormal coordinate frame. Once this frame was established, the position of the cup was measured as follows: The inclination angle was defined as the angle between the plane of the cup rim and the transverse plane of the pelvis. The anteversion angle was defined by the angle between the lateral axis of the pelvis and the projection of the normal vector to the plane of the cup rim onto the transverse plane. These angles correspond to the anatomic criteria described by Murray et al. [27].

### 2.4. 2D Analysis

The challenge of the present study was to calculate the three-dimensional anatomic angles described above, using only data from 2D fluoroscopic images. The proposed solution uses mathematical formulas calculated from the shape of the ellipse formed by the rim of the acetabular cup, as shown in Figure 3.

The JointPoint software (JointPoint, Belleair Beach, FL, USA) was used to analyze the images and perform the calculations. In accordance with the software workflow, two AP radiographic images were collected. The first image is a full pelvis view, used to establish the medial–lateral axis of the pelvis by selecting the teardrop points to create an inter-teardrop line. The brim line is also identified on the full-pelvis image (line tangent to the quadrilateral surface bisecting the radiographic teardrop (Figure 4A). The second image is centered on the acetabular component and is used to identify the ellipse created by the rim of the acetabular component, as well as the brim line, which is visible in both images (Figure 4B). By identifying the brim line on both images, the medial–lateral axis of the pelvis can be established in reference to the second image, and the anatomic inclination and anteversion angles described by Murray et al. can be calculated using the equations presented in Figure 3. 

Two orthopedic surgeons familiar with the radiographic interpretation of acetabular component placement independently analyzed the fluoroscopic hip images to measure interrater reliability, with one rater analyzing the images a second time at a 2-week interval to measure intrarater reliability. The images were presented in random order and the investigators were blind to the target position. 

### 2.5. Statistical Analysis

For each analysis of repeatability and reliability, a two-way random effect model was used to calculate the single measures, absolute agreement version of the intraclass correlation coefficient (ICC). Non-parametric 95% bootstrap confidence intervals were reported with each ICC calculation. To further assess reliability, Bland–Altman 95% limit of agreement analyses were performed. The ICC values that were >0.75 were interpreted as excellent agreement. All statistical analyses were performed with the statistical package R version 3.5.0 (R Development CoreTeam, Vienna, Austria).

## 3. Results

The range of anteversion and inclination for the radiographic images analyzed by JointPoint, as measured by the coordinate measurement system, was from 6 to 47 degrees and from 15 to 62 degrees, respectively.

Strong interrater agreement between investigators was found for images containing only the acetabular cup, with an ICC of 0.988 (95% CI 0.975–0.994) for anteversion and 0.997 (95% CI 0.991–0.999) for inclination. Only the first set of measurements was used for comparison for the investigator who completed the measurements twice. Interrater Bland–Altman limits of agreement demonstrated a range of −4 to 4 centered at about 0 degrees for anteversion and −3 to 1 centered at about −1 degrees for inclination, which are demonstrated in Figure 5 for the images containing only the acetabular cup. Strong agreement was found for interrater agreement for both inclination and anteversion parameters among images with and without femoral heads, as described in Table 1.

Comparing anteversion measurements taken with images containing only the acetabular cup with no femoral head, interpreted by the same investigator two weeks apart, an intrarater reliability ICC of 0.995 (95% CI: 0.985–0.998) was calculated. For inclination using the same images, an intrarater agreement ICC of 0.998 (95% CI: 0.994–0.999) was calculated. Bland–Altman limits of agreement for the intrarater agreement calculations were −2 to 3 degrees centered about 0 and −2 to 2 centered about 0 degrees, respectively, for anteversion and inclination. Strong intrarater agreement was similarly found for both inclination and anteversion parameters for images also containing small and large femoral heads, as shown in Table 2 for all intrarater agreement values. Bland–Altman plots demonstrating limits of agreement for the images with the acetabular component only are presented in Figure 6.

Intermethod reliability, defined as an agreement between the measurements of one investigator using the JointPoint compared to the true measurement acquired using the robotic coordinate measuring system, demonstrated strong agreement for both anteversion and inclination, with an interclass coefficient of 0.986 (95% CI: 0.972–0.993) calculated for anteversion and 0.993 (95% CI: 0.989–0.995) for inclination. Bland–Altman analysis for the intermethod agreement demonstrated limits of agreement for anteversion of −5 to 4 degrees centered about −1 degree. Limits of agreement for inclination were found to be 0 to 4 degrees, centered at about 2 degrees. Intermethod agreement remained strong for the measurements also taken with small and large femoral heads in place. Interestingly, bias was also centered around approximately two degrees for the Bland–Altman inclination measurements for images with small and large heads in place, similar to what was observed for the images only containing the acetabular cup. Full detail of the intermethod agreement measurements is contained in Table 3. Bland–Altman plots demonstrating limits of agreement for the images with the acetabular component only are presented in Figure 7.

## 4. Discussion

The most important finding of this study is the validation of the intraoperative fluoroscopic analysis software system as an accurate and reliable method for acetabular component position assessment in a pelvis model, regardless of the presence of a femoral stem and head. Further studies are necessary to validate this method in the clinical setting. Accurate placement of the acetabular component during total hip arthroplasty is critically important, as malposition of the acetabular component may contribute to prosthetic impingement, instability, eccentric wear, edge loading, aseptic loosening, osteolysis, or liner fracture [2,3,28,29,30,31,32]. Historically, acetabular component placement has been based on the “safe zone” proposed by Lewinnek et al. [4]. Traditional methods of component placement rely on identifying anatomic landmarks or upon surgeon experience, yet in the case of low volume surgeons, obese patients, and minimally invasive surgical approaches, these traditional methods for acetabular component positioning may lead to a heightened risk for malpositioning [3,33].

The use of intraoperative fluoroscopy during hip arthroplasty is purported to improve the accurate placement of the acetabular component [6,17,29,34,35]. Rathod et al. demonstrated that after a surgeon performed one hundred total hip arthroplasties using fluoroscopy, the use of fluoroscopy decreased the variability in acetabular component position for patients positioned in the supine position during a direct anterior approach compared to a posterior, unguided approach [6]. The benefits of intraoperative fluoroscopic guidance are most notable when fluoroscopy is used on the supine patient, demonstrating decreased component variability compared to the use of fluoroscopy during a lateral approach [17,29,34,35]. With the demand for primary THA expected to increase by 174% from 2005–2030 and the availability of intraoperative fluoroscopy being almost universal in developed countries, it is becoming increasingly important to implement strategies that reduce malalignment and complications associated with malalignment without significantly increasing the time in the operating room [34,35,36]. This is magnified by the fact that over a third of primary THAs are being performed in hospitals with lower volumes of these procedures, where data has shown that surgeons with lower volumes have higher rates of malalignment and complications [34,37]. While these studies have demonstrated the promise of improved acetabular component positioning with the use of intraoperative imaging and fluoroscopic assessment software, the previous literature does not discuss the accuracy or reproducibility of these analysis systems [34,35,38]. The present study is the first to validate a fluoroscopic analysis system as an accurate and reproducible method to measure acetabular component position. The results demonstrate excellent intrarater, interrater, and intermethod agreement. 

The most important clinical finding is the narrow range observed for Bland–Altman level of agreement between the 3D method and the 2D method. For the intermethod analysis, a range of approximately +/−2 to 5 degrees was observed, indicating that with high confidence, the 2D method achieved a measurement within 2–5 degrees of the intended target. This narrow range for the limits of agreement combined with the inter- and intrarater reliability results offers confidence for generalizing the expectation of accuracy across different users and among the same user across multiple measurements. While the study methodology did not identify a formal difference between the presence or absence of the femoral head, a progressive increase in the limits of agreement was observed between the images taken without a femoral head in place when compared to the images with a small and large femoral head present. It was also noted by the raters that identifying the acetabular rim was more difficult with the femoral head in place, which is one explanation for the observed increase in the limits of agreement. While specific comparative conclusions regarding the difference in accuracy between the presence or absence of the femoral head cannot be made, the accuracy and repeatability of the software measurements remained strong with the presence of the femoral head. An unexpected finding was the bias observed for the inclination angle, showing that the 2D method consistently over-estimated inclination by two degrees compared to the 3D method. This may be due to the fact that the trans-teardrop line for pelvis rotation was measured only once by the CMM at the beginning of the image acquisition process, and only once at the start of the fluoroscopic analysis system process. If those two measurements were off by two degrees, this may have produced a consistent disparity in the measurement of inclination, which is most sensitive to the rotational calibration of the image plane. 

The study is not without limitations. First, this study does not account for variability in the quality of the radiograph when applied to the clinical setting. The current study assumes a consistent radiographic plane and does not account for clinical factors such as patient body habitus or radiographic technique which have the potential to introduce error. Second, the investigators performing the image analysis were experienced in using the software and assigning an ellipse to the radiographic images. Less experienced users or varying component geometry may make ellipse alignment difficult and introduce measurement error. Third, the method was tested in nine cup positions, designed to encompass the common range of anteversion and inclination angles seen in practice. However, more cup positions could be tested to validate the method over a wider range of extreme anteversion and inclination angles or to add granularity to the validation within the tested range. 

## 5. Conclusions

The present study validates an intraoperative fluoroscopic analysis software system as an accurate and reliable method for acetabular component position assessment, regardless of the presence of a femoral stem and head. Future studies may use the clinical data produced with the use of the software to better understand accurate acetabular component positioning, which may then serve as a tool to redefine the historical “safe zone”.

## Figures and Tables

**Figure 1 medicina-58-00663-f001:**
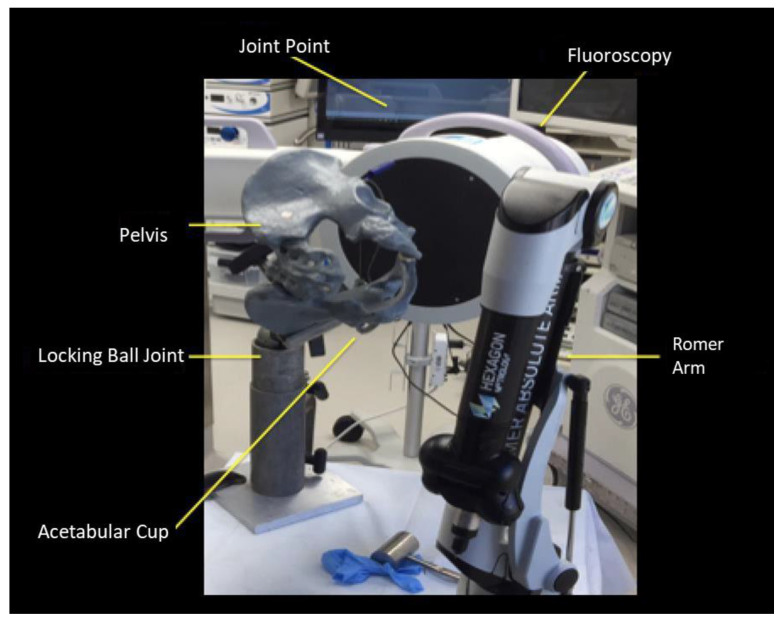
Experimental set-up including pelvis model secured with acetabular component in place, fluoroscopy unit, and the coordinate measure machine Romer Arm.

**Figure 2 medicina-58-00663-f002:**
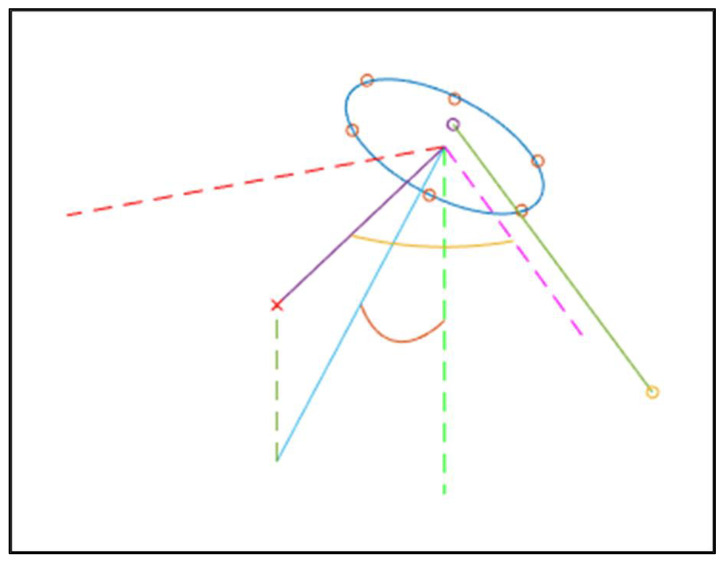
Representation of the 3D data collected using the CMM. The dashed lines represent the anatomic coordinate frame with the anterior axis (red) perpendicular to the plane of the X-ray, the lateral axis (pink) parallel to the inter-teardrop line (full green line connecting the yellow and purple points), and the inferior axis (green) mutually perpendicular to the other two axes. The red circles represent the points collected along the acetabular rim, and the blue ellipse describes the contour of the acetabular rim, calculated using a best fit circle in 3D. The light blue line represents the normal axis to the plane of the acetabular rim, and the red arc between the inferior axis (dashed green line) and the light blue line is the anatomic inclination angle described by Murray et al. The full purple line is the projection of the normal axis onto the transverse plane (plane described by the red dashed line and the pink dashed line). The yellow arc between the lateral axis (dashed pink line) and the purple line is the anatomic anteversion angle described by Murray et al.

**Figure 3 medicina-58-00663-f003:**
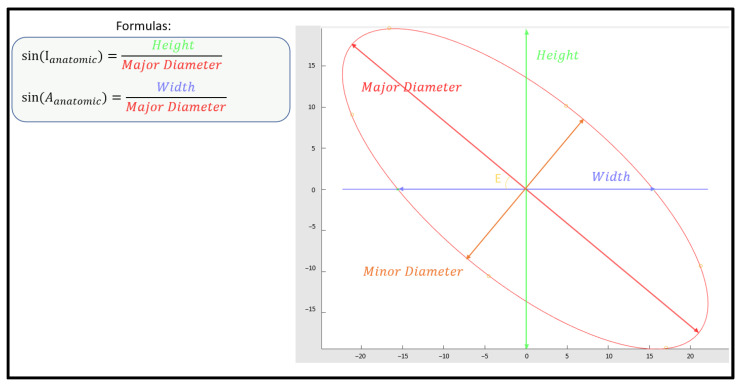
Calculation of the anatomic anteversion and inclination based on the ellipse described by the acetabular component rim. The horizontal axis is aligned with the medial/lateral axis of the pelvis using the teardrop line.

**Figure 4 medicina-58-00663-f004:**
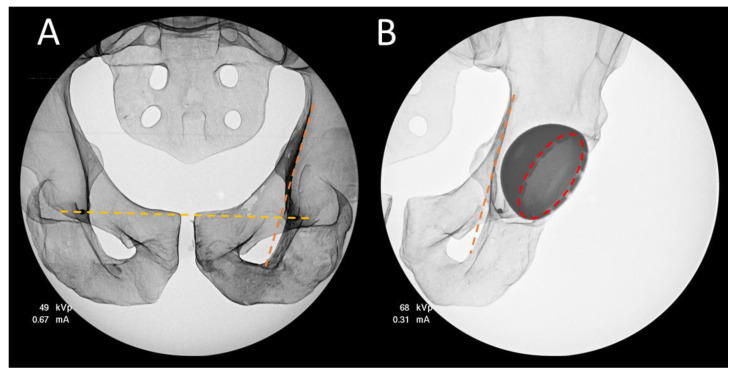
Fluoroscopic images used for 3D angle calculations. (**A**) shows the full-pelvis AP view with the teardrop line passing through the two dark circles created by the previously described steel beads, and the brim line. (**B**) Shows the acetabular cup view with the brim line and the ellipse described by the acetabular cup rim. The teardrop-brim line angle is used to identify the medial–lateral axis on the acetabular cup view.

**Figure 5 medicina-58-00663-f005:**
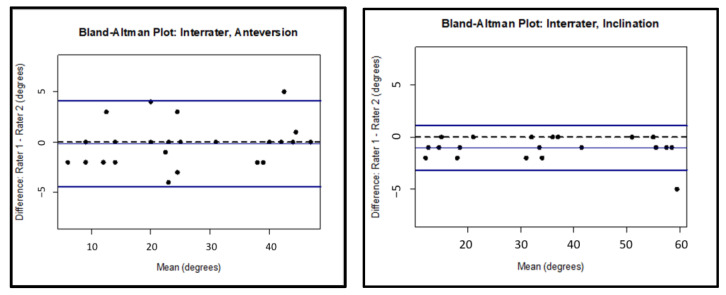
Bland–Altman limits of agreement for the interrater analysis for the cup only images.

**Figure 6 medicina-58-00663-f006:**
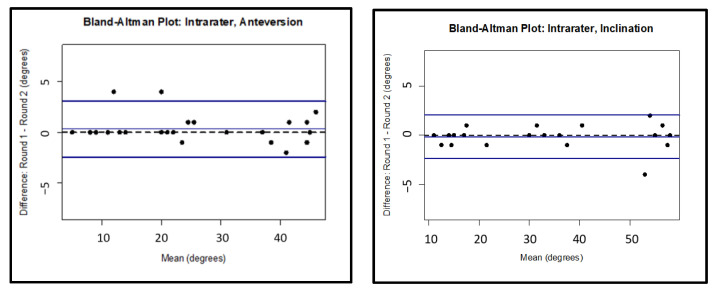
Bland–Altman limits of agreement for the intrarater analysis for the cup only images.

**Figure 7 medicina-58-00663-f007:**
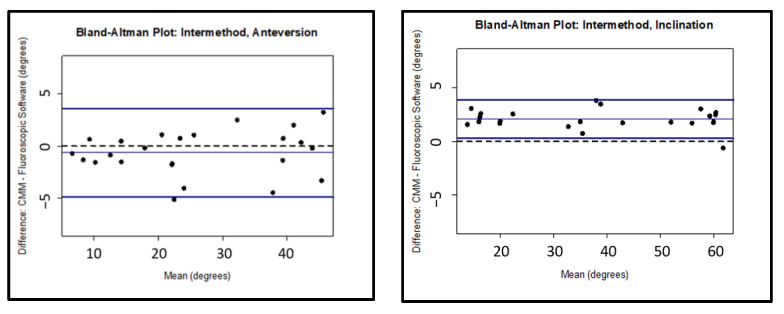
Bland–Altman limits of agreement for the intermethod analysis comparing the software analysis to the CMM analysis for the cup only images.

**Table 1 medicina-58-00663-t001:** Complete intrarater analysis including interclass correlation coefficient (ICC) and Bland–Altman bias and limits of agreement (LOA).

Intrarater							
		ICC	95% CI Lower Border	95% CI Upper Border	Bias	Lower LOA	Upper LOA
Anteversion	Cup Only	0.995	0.985	0.998	0.320	−2.432	3.072
	Cup plus Small Head	0.983	0.952	0.995	−1.500	−5.295	2.295
	Cup plus Big Head	0.985	0.971	0.993	−0.722	−5.210	3.766
Inclination	Cup Only	0.998	0.994	0.999	−0.160	−2.372	2.052
	Cup plus Small Head	0.999	0.998	1.000	−0.188	−1.856	1.481
	Cup plus Big Head	0.999	0.997	0.999	−0.056	−1.801	1.690

**Table 2 medicina-58-00663-t002:** Complete interrater analysis including interclass correlation coefficient (ICC) and Bland–Altman bias and limits of agreement (LOA).

Interrater							
		ICC	95% CI Lower Border	95% CI Upper Border	Bias	Lower LOA	Upper LOA
Anteversion	Cup Only	0.988	0.976	0.995	−0.160	−4.429	4.109
	Cup plus Small Head	0.941	0.868	0.976	−3.625	−8.834	1.584
	Cup plus Big Head	0.940	0.871	0.974	−3.056	−10.459	4.348
Inclination	Cup Only	0.997	0.991	0.998	−1.040	−3.160	1.080
	Cup plus Small Head	0.996	0.989	0.998	−1.438	−3.499	0.624
	Cup plus Big Head	0.996	0.988	0.999	−0.833	−3.512	1.846

**Table 3 medicina-58-00663-t003:** Complete intermethod analysis comparing the software analysis to the CMM analysis including interclass correlation coefficient (ICC) and Bland–Altman bias and limits of agreement (LOA).

Intermethod							
		ICC	95% CI Lower Border	95% CI Upper Border	Bias	Lower LOA	Upper LOA
Anteversion	Cup Only	0.986	0.969	0.993	−0.631	−4.886	3.625
	Cup plus Small Head	0.978	0.941	0.994	−0.994	−6.094	4.106
	Cup plus Big Head	0.966	0.912	0.989	−1.271	−7.791	5.249
Inclination	Cup Only	0.993	0.988	0.995	2.083	0.282	3.885
	Cup plus Small Head	0.993	0.989	0.995	2.133	0.182	4.084
	Cup plus Big Head	0.988	0.982	0.992	2.438	−0.147	5.022

## Data Availability

Not applicable.

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
