# Peer review of "Accuracy and Reliability of Software Navigation for Acetabular Component Placement in THA: An In Vitro Validation Study"

_medicina, 2022, doi:10.3390/medicina58050663_

Round 1

Reviewer 1 Report

The approach is original and simple that makes this article very interesting with a great interest for publication.

Several issues must be addressed:

  • Definitions must be given and specific.
  • Figures must be revised to be easier to understand and guide the reader in a step by step description
  • Method must be more structured to differentiate, same for the results
  • Information must be given on the clinical interest, notably because readers do not know in which extend the fluoroscopy is used in a routine practice by surgeons. This is very important for the clinical relevancy of the article as the solution could be very interesting but its use very poor.

Abstract:

It is not clear what is the imaging that is used for the first method as it is specified for the second one.

The collection of the data must be described prior to the method of measurements.

“level of agreement” would better say “limit of agreement”

Introduction

“This method has the benefit of offering immediate feedback to the surgeon and enables intraoperative adjustment to obtain ideal component position.”: it is already looks like a conclusion. Please remove as it is not proved at this moment.

“This will provide…”: not demonstrated yet. Must be reformulate like a hypothesis.

Methods

Globally, the methods must be more structured with a distinction between 1) the referential in which the measurements are done, 2) a description of the collected data that allows surgeon measurement, 3) the description of the intraoperative automatic system (principles and practice application in this specific case).

Figures must follow this plane and be more specific and of higher quality

“the anterior pelvic plane”: please specify. There are plenty description of anterior pelvic plane in the literature.

“teardrop line was physically marked with small steel beads”: does it mean manually? Who did this marking?

A figure must be done here to support visually the various geometric landmarks.

“as measured by 110 JointPoint to » JointPoint must be described before.

“The anterior axis was defined as perpendicular to the plane of the X-ray”: this description does not allow any reader to reproduce the experience. Please be more specific.

Must be more specific about what is expected from the measurements done with and without the femoral head, that is not clear.

Results

Please structure this section better: 1) ICC between surgeons that evaluate the concordance, 2) comparison between 2 methods (robotic/automatic vs handmade by the expert surgeons) that is the place for Bland Altman).

Please do not use decimales for version and inclination

“Bland-Altman plots demonstrating limits of agreement the images with the acetabular component only are presented in Figure 5.” Not clear, please reformulate.

Discussion

Well written but again, 2 or 3 sections would help the reader to better track the information.

Information must be given on the clinical interest, notably because readers do not know in which extend the fluoroscopy is used in a routine practice by surgeons. This is very important for the clinical relevancy of the article as the solution could be very interesting but its use very poor.

Reviewer 2 Report

The manuscript under review is an interesting study on the use of a 3-Dimensional imaging technique in THA. The study aimed to evaluate the accuracy and validate this new method on pelvis model. The tested cup positions are nine. 3-D models and analysis are well explained, the figures are good. The discussion and conclusions are consistent with the results. However, I have some minor comments:

  • In the “methods” section, lines 111-113, the authors say “The nine tested cup positions were 111 chosen to validate the accuracy…”. I suggest adding the explanation why these nine cup positions were chosen.
  • In the discussion, the authors stated that “The most important finding of this study is the validation of the intraoperative fluoroscopic analysis software system as an accurate and reliable method for acetabular component position assessment, regardless of the presence of a femoral stem and head”. I think it is premature to say that this study validates the proposed 3-D analysis since it is conducted only in pelvis models and do not consider the clinical variability. I suggest the Authors re-writing this sentence. Further clinical studies as certainly needed to validate this technique.
  • When exposing the limitations of the study, the authors did not mention the limited number of tested cup positions (9). I suggest adding this limit.
  • The reference list should be implemented with the following papers: 10.1117/1.JMI.5.2.021205; 3389/fped.2020.00207; 10.3928/01477447-20111021-08.

Round 2

Reviewer 1 Report

The authors have improved the manuscript. 

As proposed in my first review, a more structured discussion could make it more readable but it can be published in that form. 

Authors must be congratulated for this interesting work